# Oral Intake of Inosine 5′-Monophosphate in Mice Promotes the Absorption of Exogenous Fatty Acids and Their Conversion into Triglycerides though Enhancing the Phosphorylation of Adenosine 5′-Monophosphate-Activated Protein Kinase in the Liver, Leading to Lipohyperplasia

**DOI:** 10.3390/ijms241914588

**Published:** 2023-09-26

**Authors:** Bin Zhang, Yang Xu, Jinyan Liu, Chongming Wu, Xiaohong Zhao, Lidong Zhou, Yong Xie

**Affiliations:** Institute of Medicinal Plant Development, Chinese Academy of Medical Science & Peking Union Medical College, Malianwa North Road No. 151, Haidian District, Beijing 100193, China; bzhang@implad.ac.cn (B.Z.); yxu@implad.ac.cn (Y.X.); 15666252538@163.com (J.L.); chongmingwu@163.com (C.W.); xhzhao@implad.ac.cn (X.Z.); ldzhou@implad.ac.cn (L.Z.)

**Keywords:** inosine 5′-monophosphate (IMP), AMPK, acetyl-CoA, ACC, lipohyperplasia

## Abstract

Inosine 5′-monophoaphate (IMP) is a food additive that promotes serious lipohyperplasia in the liver of C57/KsJ-*db*/*db* (*db*/*db*) mice. Thus, IMP taken orally by healthy mice might also damage their health. To date, how IMP affects health after being taken by healthy animals is still unclear. Therefore, we investigated the health of C57BL/6J mice affected by IMP intake. Our data revealed that C57BL/6J mice administered 255 μM IMP daily via oral gavage for 4 months caused hyperlipidemia and an increase in body fat rate. The expressions of acetyl-CoA carboxylase 1 (ACC1) and phosphorylated acetyl-CoA carboxylase 2 (ACC2) in hepatocytes increased though the administration of IMP, promoting the phosphorylation of adenosine 5′-monophosphate-activated protein kinase (AMPK). The conversion of acetyl-CoA into triglycerides (TGs) was promoted by ACC1. These TGs were transported from the hepatocytes to avoid the development of non-alcoholic fatty liver disease (NAFLD), causing a deficiency of acetyl-CoA in the liver, and then, the increased phosphorylated ACC2 promoted the cytoplasm fatty acids entering the mitochondria and conversion into acetyl-CoA through the fatty acid β-oxidation pathway, causing a deficiency in fatty acids. Therefore, the liver showed enhanced absorption of exogenous fatty acids, which were converted into TGs, causing lipohyperplasia. In conclusion, an excessive IMP intake promotes metabolic dysfunction in adipose tissue.

## 1. Introduction

Inosine 5′-monophoaphate (IMP) is a purine nucleotide that is necessary for organisms and allows animals to sense umami flavors [1,2,3]. By taking advantage of the IMP–disodium salt complex (IMP-Na_2_), the production of IMP has been achieved at an industrial level [4]. As IMP-Na_2_ has demonstrated very low toxicity to mammals [5], the Codex Alimentarius Commission (CAC) approved IMP-Na_2_ as a food additive and considered it unnecessary to stipulate the acceptable daily intake (ADI) for human beings (FAO/WHO, 1994). The results of an in vitro assay revealed that, while IMP is absorbed by mammalian cells, it cannot be degraded in a timely manner, resulting in its accumulation in cells [6]; moreover, IMP can be converted into other purine nucleotides [7]. Thus, the intake of IMP via the diets of mammals may produce excessive purine nucleotides in their bodies.

In a previous study, it was identified that C57/KsJ-db/*db* (*db*/*db*) mice administered 50 mg/(kg *m*_b_·d) of IMP-Na_2_ via oral gavage for 8 weeks had lipohyperplasia of the liver, causing non-alcoholic fatty liver cirrhosis (NAFLC) [8], which easily evolved into hepatocellular carcinoma, resulting in death [9]. Due to leptin receptor gene defects, *db*/*db* mice had severe leptin resistance, resulting in spontaneous metabolic syndromes, such as NAFLD and type-II diabetes [10]. Therefore, IMP should cause NAFLC in humans with severe leptin resistance. Based on the Body Surface Area-Based Equivalent Dose Calculation Rule between mice and humans [11,12], we can assume that an intake of 285 mg/d of the IMP-Na_2_ via the diet of humans with severe leptin resistance and a body weight of 70 kg may induce NAFLC. In addition, an excessive glutamine intake of C57BL/6J mice via their drinking water induced excessive levels of IMP and UA in the body, causing hypothalamic inflammation, leptin resistance, and oxidative stress, which together caused an excessive caloric intake, promoting metabolic syndromes, such as adiposity and NAFLD [13]. These findings revealed that oral intake or metabolic disorders cause IMP accumulation in mammals, which promotes metabolic syndromes. At present, they ways in which IMP—added to the diet to increase its umami flavor—affects the health of healthy mammals is unclear. Modern-day humans have developed a habit of eating regularly in order to avoid the metabolic syndromes caused by disrupted food intervals [14,15]. Judging from the eating habits of modern-day humans, IMP is usually ingested through food and food additives during meals; thus, it can be assumed that the metabolic dysfunctions in C57/BL6J mice induced by the ingestion of IMP via oral gavage every day are consistent with those of an excessive intake of IMP from food by humans. The investigation of the metabolic functions of 4-month-old C57BL/6J mice that were gavaged with various doses of IMP at 24-h intervals for 4 months was conducted in this study. This investigation elucidated the harmful dose of IMP-Na_2_ for C57/BL6J mice and the novel mechanisms of IMP that induce hyperlipidemia and body fat hyperplasia, as well as promote inflammation in mice. Our data provide novel insights into the metabolic dysfunctions of the adipose tissue of mammals, as well as the food safety of IMP-Na_2_.

## 2. Results

### 2.1. Intake of IMP in C57BL/6J Mice Induces Lipohyperplasia and Inflammation

Twenty C57/BL6J mice were randomly divided into four groups. The mice in the control group were normally fed, and the mice in the other three groups were gavaged with 10, 50, and 100 mg/kg of IMP-Na_2_, respectively (referred as the low-, middle-, and high-dose groups, respectively). During the experimental period, the mice in each IMP-Na_2_ treatment group did not have an excessive food intake or experience weight gain in comparison to the control group (Figure 1A). At the end of the experiment, only the body fat rate of the mice in the high-dose group increased, by approximately 55% in comparison to the control group (Figure 1B). The levels of TGs, non-esterified fatty acids (NEFAs), and total cholesterol (TC) in the liver tissue did not show a significant change in comparison to the control group (Figure 1C). The serum TG content of the mice in the high-dose group increased by approximately 2 times; the serum TC content of the mice in the low-dose group was reduced by approximately 50%; and the levels of serum low-density lipoprotein cholesterol (LDL), high-density lipoprotein cholesterol (HDL), and uric acid (UA) of the mice in each group treated with the IMP presented no change in comparison to the control group (Figure 1D).

The serum tumor necrosis factor-α (TNF-α) content of the mice in each of the IMP-Na_2_ treatment groups was higher than that of the mice in the control group, and the content depended on the dose of IMP. In contrast, the contents of interleukin-1β (IL-1β), interleukin-6 (IL-6), alkaline phosphatase (ALP), aspartate aminotransferase (AST), alanine aminotransferase (ALT), and lactate dehydrogenase (LDH) did not change (Figure 2A). These results revealed that an intake of more than 10 mg/(kg *m*_b_·d) IMP-Na_2_ could induce inflammation, and inflammation intensity is dependent on the intake of IMP. Even the intake of IMP-Na_2_ up to 100 mg/(kg *m*_b_·d) in the mice could not promote IL-1β or IL-6 increasing, which suggests that their organs were not still injured [16].

The contents of TG, NEFA, and TC in the liver tissue and AST, LDL, and LAT in the serum of some individual members in each administration group were significantly higher than those of the members in the same group (Figure 1C and Figure 2A). Their liver tissue sections were stained with Oil Red O, and results indicated that NAFLD occurred (Figure 2B). Due to the interindividual variability in metabolism, some members of the control C57/BL6J mice population exhibited NAFLD [17]; therefore, the intake of IMP via gavage was not the cause of NAFLD in these C57/BL6J mice. Our data revealed that an intake of 50 mg/(kg *m*_b_·d) IMP-Na_2_ in healthy mice did not induce any metabolic syndrome, while an intake dose of IMP-Na_2_ up to 100 mg/(kg *m*_b_·d) induced lipohyperplasia, presenting as raised body fat rate and hyperlipidemia.

### 2.2. Adenosine 5′-Monophosphate-Activated Protein Kinase (AMPK) Is the Unique Target for IMP-Regulated TG Metabolism

IMP or purine nucleotides such as adenosine 5′-monophoaphate (AMP) and adenylosuccinate form a complex with the γ-subunit of adenosine 5′-monophoaphate activited protein kinase (AMPK) to promote the phosphorylation of the α-subunit of AMPK, leading to TGs increasing in vivo; in contrast, this leads to TGs decreasing in vitro [8,18]. To demonstrate that AMPK is the unique target for the IMP regulation of lipid metabolism, we measured the lipid-lowering activity of IMP in the adipose-accumulating HepG2 cells induced by oleic acid (OA) with and without 6-[4-(2-piperidine-1-ethoxy) phenyl]-3-pyridine-4-pyrazolazo [1,5-a] pyrimidine (dorsomorphin), which is a highly effective AMPK inhibitor [19]. Results revealed that when dorsomorphin was absent, the lipid-lowering activity displayed by IMP was basically the same as that of lovastatin; in contrast, when dorsomorphin was present, the lipid-lowering function of IMP was lost (Figure 3). AMPK is not the target of lovastatin, which achieves a lipid-lowing effect targeting hydroxymethylglutaryl-CoA (HMG-CoA) reductase [20]; therefore, the inactivation of AMPK did not alter the lipid-lowering activity of lovastatin. Our data demonstrated that IMP binds to AMPK to change the expression level of key proteins in the TGs metabolic pathway, causing TGs accumulation or disappearance. AMPK is the unique target for IMP-regulated TGs metabolism.

### 2.3. Intake of IMP in Healthy Mice Promoted Acetyl-CoA Regeneration though Enhancing Phosphorylation of AMPK

The comparison of the protein expression level in the hepatocytes of mice in the low-, middle-, and high-dose groups with that of the mice in the control group revealed that phosphorylated AMPK increased by about 0.8, 0.9, and 1.2 times, respectively; acetyl-CoA carboxylase 1 (ACC1) increased by about 0.5, 2.5, and 2.8 times, respectively; adipose triglyceride lipase (ATGL) in the hepatocytes of the mice in all of the IMP-treated groups increased by about 1.5 times; acetyl-CoA carboxylase 2 (ACC2) as well as phosphorylated ACC2in the hepatocytes of the mice in the high-dose group increased by about 0.5 times, respectively (Figure 4). ACC1 exists in the cytoplasm of hepatocytes and is a rate-limiting enzyme in the acetyl-CoA-to-fatty acid pathway [21,22]. Fatty acids in hepatocytes are majorly converted to TGs through the diglyceride pathway [23]. Thus, increasing ACC1 expression promoted the conversion of acetyl-CoA to TGs in hepatocytes. ACC2 exists in the outer membrane of mitochondria, and increasing the phosphorylated ACC2 in hepatocytes promotes the fatty acids in the cytoplasm being transported into the mitochondria and then being converted into acetyl-CoA by the fatty acid β-oxidation pathway [24]. The increasing expression of ATGL promotes the degradation of intracellular TGs into fatty acids and glycerol [25].

Our data revealed that the activities of the AMPK-ACC1 and AMPK-ACC2 pathways increased depending on the intake of IMP; however, the activity of the AMPK-ATGL pathway increased independently of the intake dose of IMP. Therefore, the intake of IMP should cause changes in the content of TG, NEFA, or acetyl-CoA in vivo; in fact, the contents of TG and NEFA in the liver of the mice in the low-, middle-, and high-dose groups did not change in comparison to those of the mice in the control group (Figure 1C). Additionally, while the content of acetyl-CoA in the liver or serum was lower than that of the mice in the control group, the content of acetyl-CoA increased depending on the intake dose of IMP (Figure 5). These results suggest that the intake of IMP promoted TG synthesis by promoting the regeneration of acetyl-CoA in vivo, as the mice in the high-dose group showed body fat accumulation and hyperlipidemia (Figure 1D).

### 2.4. IMP Reduced Activity of Complement System to Induce Inflammation

In order to elucidate the newly discovered mechanism of IMP promoting lipohyperplasia and inflammation in mice, identification of the differentially expressed proteins in the liver of the mice in the high-dose group vs. those of the mice in the control group was carried out using tandem mass tag (TMT)-based proteomics analysis. The liver tissues of three mice in the high-dose group and three mice in the control group were randomly selected as test samples. The MS/MS data of proteins extracted from each sample after enzymatic hydrolysis and TMT modification met the quality requirements for the classification of all proteins in hepatocytes and quantitative analysis of each protein (Appendix A). The distribution characteristics of peptide sequences identified using MS/MS data were in accordance with the law of protein cleavage (Appendix A), and the proteins not successfully identified accounted for 3.98% of the total proteins of hepatocytes (Appendix A). A total of 4955 proteins were identified in the livers of the control group and the high-dose group, and 4769 proteins were determined (Appendix A). The quantitative proteins in the liver tissue samples of healthy mice were used as control, and the expressions of 4 proteins were increased (Table 1) and those of 23 proteins were decreased (Table 2) in the liver tissue of the high-dose mice. The increased expression of elongation of very long chain fatty acids protein 5 (Q8BHI7, ELOV5) suggested that the conversion of acetyl-CoA into fatty acids was promoted by the AMPK-ACC1 pathway [26]. The increased expression of abhydrolase domain containing 2-acylglycerol lipase (Q9QXM0, ABHD2) suggested that the AMPK-ATGL pathway was activated [25]. Except for those two proteins, TMT-based proteomics analysis did not find significant changes in the expression of proteins regulating adipose metabolism.

The results of gene ontology (GO) annotation [27] for these 27 proteins suggested that eight biological functions involved in the immune system in the liver were significantly changed by IMP (Figure 6A). The phenotype of the mice in the high-dose group suggested that IMP induced significant changes in the function of the hepatocyte immune process. The expression of up-regulated hepcidin (Q9EQ21) as well as expressions of down-regulated proteins complement factor H (P06909, CFAH), complement component C4-B (P01029, C4B), complement component C8 *γ* chain (Q8VCG4, C8G), complement component C8*α* chain (Q8K182, C8A), and Haptoglobin (Q61646, HPT) were directly related to the changes in immune processes, causing inflammation (Figure 6B). The results of signal pathway enrichment using the Kyoto Encyclopedia of Genes and Genomes (KEGG) mapping tool [28] revealed that the change in the complement signaling pathway in hepatocytes was the cause of the immune process changes (Figure 6C). The expression changes in CFAH, C4B, complement component C5 (P06684, C5), C8G, C8A, and complement component C9 (P06683, C9) were directly involved in regulating the changes in the complement system [29,30] (Figure 6D). The proteins C4B, C5, C8G, C8A, and C9 constitute signaling pathways that regulate the generation of complement component C5α (C5α) and membrance attack complex (MAC) in the complement system [30] (Figure 6E). The expression levels of C4B, C5, C8G, C8A, and C9 were all decreased by IMP, resulting in the inhibition of the formation of C5α and MAC complexes, which caused the function of the complement system to weaken, promoting inflammation. The mechanism of IMP binding to AMPK to cause the complement system to weaken could not be elucidated based on our data.

## 3. Discussion

Results of this study revealed that phenotypes of the C57/BL6J mice induced by IMP-Na_2_ at 24-h intervals were obviously different from those of the intake of glutamine from drinking water [13]. Our data suggest that the intake of IMP by healthy mice promoted the phosphorylation of the AMPK-activated AMPK-ACC1, AMPK-ACC2, and AMPK-ATGL pathways in hepatocytes, causing TGs accumulation in vivo; therefore, elucidating the mechanism of the periodic intake of IMP-induced lipohyperplasia is important to understand the purine nucleotides metabolic disorder relating to TGs metabolic disorder, and to detail the food safety of IMP-Na_2_.

If the diglycerol pathway in the hepatocytes of the mice in the low-dose group was completely inhibited by the AMPK-ATGL pathway, the fatty acids converted by acetyl-CoA would accumulate in the hepatocytes. In fact, the contents of NAFAs and TGs in the liver of the mice in the low-dose group did not change (Figure 1C). Therefore, even though the activity of the AMPK-ACC1 pathway was similar to that of the mice in the high-dose group, the diglycerol pathway could not be completely inhibited. Once the AMPK-ACC1 pathway was activated by IMP, a metabolic pathway was formed in the hepatocytes to promote the conversion of acetyl-CoA into TGs and its transport outside the liver. As the NEFAs and TGs contents remained unchanged, the liver cells simply consumed their own acetyl-CoA, and a severe deficiency of acetyl-CoA inevitably inhibited TC generation. Therefore, the serum TC content of the mice in the low-dose group decreased (Figure 1D). With the increase in the IMP dose, the activity of the AMPK-ACC2 pathway was enhanced, and acetyl-CoA converted from the absorption of exogenous fatty acids by hepatocytes was increased. The content of acetyl-CoA in mice increased, and acetyl-CoA for TC synthesis was no longer scarce. It was observed that the content of TC in the serum of mice in the middle- and high-dose groups was unchanged compared with those in the control group. Thus, the AMPK-ATGL pathway does not contribute to the inhibition of hyperlipidemia in mice, suggesting that the activation of the AMPK-ACC1 and AMPK-ACC2 pathways is a necessary condition for the mouse liver to enhance the absorption of exogenous fatty acids, which are then converted into TG and transferred outside the liver.

The mechanism of periodic intake of IMP-induced body fat accumulation and hyperlipidemia in healthy mice, elucidated from our data, is shown in Figure 7. IMP was absorbed by the small intestine and entered the hepatocytes through blood circulation. Subsequently, the phosphorylation of AMPK was promoted, resulting in the activation of the AMPK-ACC1 pathway, AMPK-ACC2 pathway, and AMPK-ATGL pathway. The AMPK-ACC1 pathway promoted the conversion of acetyl-CoA into TG in the cytoplasm. The TGs content of the liver tissue remained unchanged (Figure 1C), indicating that the AMPK-ATGL pathway could not completely inhibit the conversion of acetyl-CoA into TG in the cytoplasm, and excess TGs was transported outside the hepatocytes, resulting in the deficiency of acetyl-CoA in hepatocytes. At the same time, the activated AMPK-ACC2 pathway promoted the transport of cytoplasmic fatty acids into the mitochondria through the fatty acid β-oxidation pathway to generate acetyl-CoA, and then it transported acetyl-CoA to the cytoplasm, which led to a deficiency of fatty acids in the hepatocytes. Thus, IMP forced the hepatocytes to enhance the absorption of external fatty acids and convert them into TGs, and then it transported the TGs outside of the hepatocytes to inhibit NAFLD. Because the leptin resistance level is dependent on IMP content in healthy mice [13], once these TGs could not be degraded in a timely manner, hyperlipidemia and body fat rate increased. The occurrence of hyperlipidemia in mice depends on whether the TGs secreted by the liver, induced by IMP, exceeds its metabolic limit. Our data confirmed that the intake of 255 μM/d IMP in mice caused the inability of TGs that was transported outside of the liver to be metabolized by other organs, resulting in hyperlipidemia and an increase in body fat rate.

In the case of the periodic intake of IMP in *db/db* mice, IMP promoted AMPK phosphorylation to induce phosphorylated ACC2 expression, increasing the expressions of ACC1 and ATGL, with no significant change in hepatocytes [8]. The increased phosphorylated ACC2 promoted the conversion of fatty acids in hepatocytes into acetyl-CoA, causing fatty acid deficiency in the liver. Thus, the liver enhanced the absorption of fatty acids from the peripheral blood to promote TGs generation, causing lipohyperplasia [8]. The results of the in vivo assay revealed that increasing the level of phosphorylated ACC2 is the requirement to promote the absorption of exogenous fatty acids and convert them into TGs synthesis in the liver to cause lipohyperplasia. In addition, the results of the in vitro assay revealed that purine nucleotides, though promoting AMPK phosphorylation, increased the expression of phosphorylated ACC2 and ATGL, while there was no significant change in the expressions of ACC1 [8,18]. The phosphorylated ACC2 also caused an intracellular deficiency in fatty acids. Because the cell culture medium did not contain additional fatty acids, the fatty acids absorbed from the outside could not maintain the content of acetyl-CoA in the cells, so the expression of ATGL increased to promote lipolysis to generate fatty acids and convert them into acetyl-CoA. Therefore, IMP showed a high activity of lipid lowering in vitro.

Hyperlipidemia is one of the factors contributing to metabolic syndromes such as obesity and atherosclerosis [31]. Except for familial metabolic deficiency, eating disorders are the main reasons promoting the onset of hyperlipidemia at a younger age [31,32]. Hyperlipidemia in this age group should not be caused by aging. Based on the age relation between mice and humans, 4–8-month-old mice are equivalent to 25–35-year-old humans; young groups who have not yet reached middle age [33]. The mechanism of lipid metabolism disorder induced by IMP in young mice (Figure 7) should be consistent with the pathogenesis of hyperlipidemia in young humans. Therefore, the intake of a high-purine diet in humans should promote the onset of hyperlipidemia at a younger age.

In addition, our data revealed that mice in the low- and medium-dose groups did not develop any metabolic syndromes but did develop inflammation, which was caused by the weakening of the complement system in the liver. Because lipohyperplasia in mammal bodies may induce meta-inflammation [34], the inflammation in mice in the high-dose group included inflammation caused by the weakening of the complement system in the liver and meta-inflammation; however, the combined action of these two kinds of inflammation did not harm the organs. Thus, it can be considered that the inflammation induced via the periodic intake of IMP is harmless to health.

## 4. Materials and Methods

### 4.1. Mice Feeding, IMP Treatment, and Physiological Data Determination

Twenty C57/BL6J mice (male; weight, 25–30 g; age, 16 weeks) from Vital River Laboratories (Qualified Certificate No. SCXK Jing 2019-0010, Beijing, China) were randomly divided into four groups. Five mice in each group were housed in a cage and in a specific-pathogen-free environment, which was maintained at a temperature of 22 ± 2.0 °C and relative humidity of 50–70%, with a 12h of light/dark cycle and free access to diet and purified water. The SPF Biotechnology Co., Ltd. Beijing, China, provided a standard diet, which was designed to meet all the nutritional requirements of healthy mice.

After the mice were fed for one week, mice in one group were fed normally as control, and mice in the other three groups were gavaged with 10, 50, and 100 mg/kg of IMP-Na_2_, respectively. The IMP treatments were administered once every 24 h. The drug treatments were carried out for 4 months. We measured the body weight of the mice at intervals of one month. At the end of the IMP-Na_2_ treatment, the of body fat rate of the mice was measured using a body composition analyzer (Bruker LF50/90, Saarbrücken, Germany). After the euthanasia of the mice, the eyeball blood was taken to measure the content of TG, TC, HDL, and LDL in each group using an automatic biochemical analyzer (Biobase BK-400, Jinan, China). To detect adipose accumulation in the liver, frozen sections were rinsed with distilled water, stained with 0.2% Oil Red O (Sigma-Aldrich, Shanghai, China) and 60% 2-propanol (Sigma-Aldrich, Shanghai, China) for 10 min at 37 °C, and then rinsed with distilled water. The liver tissues were observed using a microscope (Leica DM4000B, Benshein, Germany). The acetyl-CoA content in the serum and liver were determined following the methods of the published article [8]. Corresponding assay kits (Zhongsheng Beikong Biotechnology, Beijing, China) were used to determine the content of AST, ALP, ALT, LDH, TNF-α, IF-6, and IF-β in the liver, as well as the content of UA in the serum.

### 4.2. Western Blotting Analysis

The liver tissue of each group was lysed with RIPA buffer containing protease and phosphatase inhibitor cocktail for 30 min at 4 °C. Protein concentrations were determined using a BCA protein estimation kit (Biodragon, Beijing, China). Equal amount of proteins were separated using SDS-PAGE and transferred to PVDF membranes (Merck Millipore Ltd., Darmstadt, Germany). The membranes were then incubated in blocking buffer containing 5% non-fat milk for 2 h and probed with primary antibody overnight at 4 °C. The membranes were then washed with the TBS-T buffer three times and incubated at room temperature for 2 h with the secondary antibody that was conjugated with horseradish peroxidase. The membranes were washed three more times, and then they went through immunodetection with the promote chemiluminescent science kit (Biodragon, Beijing, China). The levels of protein expression were quantified using Image J software (v. 1.8.0, National Institutes of Health, Bethesda, MD, USA) and normalized to the relative internal standards. All the antibodies used for Westen blotting analysis were purchased from the Protein Thec., Inc. (Manchester, UK).

### 4.3. In Vitro Assay of Lipid-Lowering Effect of IMP with and without AMPK Inhibitor

The HepG2 cell line was provided by the National Infrastructure of Cell Line Resource (Beijing, China). Lovastatin (99%), IMP (99.5%), and dorsonmorphin [19] were purchased from Sigma-Aldrich company (Shanghai, China). HepG2 cells were cultured in Dulbecco’s modified Eagle’s medium (DMEM, GE Healthcare Life Sciences, South Logan, Ulah, USA) supplemented with 10% fetal bovine serum (Life technologies, Auckland, New Zealand) and 1% penicillin/streptomycin (Biodragon, Beijing, China) in a humidified atmosphere of 5% CO_2_ and 37 °C. The cells were seeded in 96-well plates with approximately 1 × 10^4^ cells in each well. After the cells were totally attached, the cells were divided into two groups. We added 0.1% (*v*/*v*) DMSO to one group and 10 μM AMPK inhibitor to the other, and both were cultured for 6 h. Then, a DMEM medium containing 100 μM OA and a DMEM medium containing 10 μM lovastatin and 100 μM OA were used as the model and the positive control, respectively. The drug treatment of the HepG2 cells was carried out using DMEM medium containing 100 μM OA and 10 μM IMP for 24 h. The HepG2 cells were cultured using DMEM solution for 24 h as a control. After the drug treatment, the cells in each well were washed with PBS and fixed with 4% paraformaldehyde for at least 30 min. After the paraformaldehyde was discarded, the HepG2 cells in each hole were stained using 100 μL of Bodipy 493/503 solution (2 μM) at 37 °C for 30 min without light. We discarded the dye and rinsed each hole with PBS three times. The cells were observed and photographed using an inverted fluorescence microscope. The fluorescence value was scanned into a gray value using Image J software (v. 1.8.0, National Institutes of Health, Bethesda, MD, USA) and normalized to the relative internal standards.

### 4.4. TMT-Based Proteomics Analyses

The liver tissues of mice from the control group and the high-dose group, respectively, were frozen and ground with liquid nitrogen into powder, and then transferred to a 5 mL centrifuge tube. Subsequently, we carried out the extraction of the total proteins, digestion using trypsin, labeling the TMT peptides, and LC-MS/MS analysis for the peptides following the reported methods [35].

The MS/MS data were processed with the Maxquant Search Engine (v.1.5.2.8, Max Planck Institute of Biochemistry, Martinsried, Germany). The mass spectra were searched in the UniProt database concatenated with the reverse decoy database. To identify differentially expressed proteins, relative protein expression was compared between the control group and the high-dose group. The two-sample equal-variance *t*-test was utilized. The criteria used to define differentially abundant proteins were based on *p*< 0.05 and fold change (FC) > 1.2.

### 4.5. Bioinformatics Analyses

The contents of bioinformatic analysis include GO annotation and online KEGG annotation. Classifications of the biological function for the differentially expressed proteins induced by IMP were determined using the GO annotation based on the UniProt-GO database (http://www.ebi.ac.uk/GOA/ (accessed on 22 June 2022)) [27]. The signal pathways involving these proteins were determined using the KEGG mapping tool based on their database (http://www.genome.jp/kegg/ (accessed on 22 June 2022)) [28].

### 4.6. Statistical Analyses

The results are expressed as mean ± standard deviation (SD). The data statistical analysis was carried out using the GraphPad Prism 5.0 software (San Diego, CA, USA). The statistical significance of the groups’ differences was analyzed with one-way ANOVA followed by Tukey’s test or the Newman–Keuls test. *p*< 0.05 was considered statistically significant.

## 5. Conclusions

In conclusion, a IMP-Na_2_ intake greater than 100 mg/(kg *m*_b_·d) in C57/BL6J mice will promote the onset of hyperlipidemia and adipose accumulation at a younger age. The IMP anion is promoted by increasing the activities of the AMPK-ACC1 and AMPK-ACC2 pathways to enhance the absorption of exogenous fatty acids and their conversion into TG in the liver, leading to adipose metabolic dysfunctions.

## Figures and Tables

**Figure 1 ijms-24-14588-f001:**
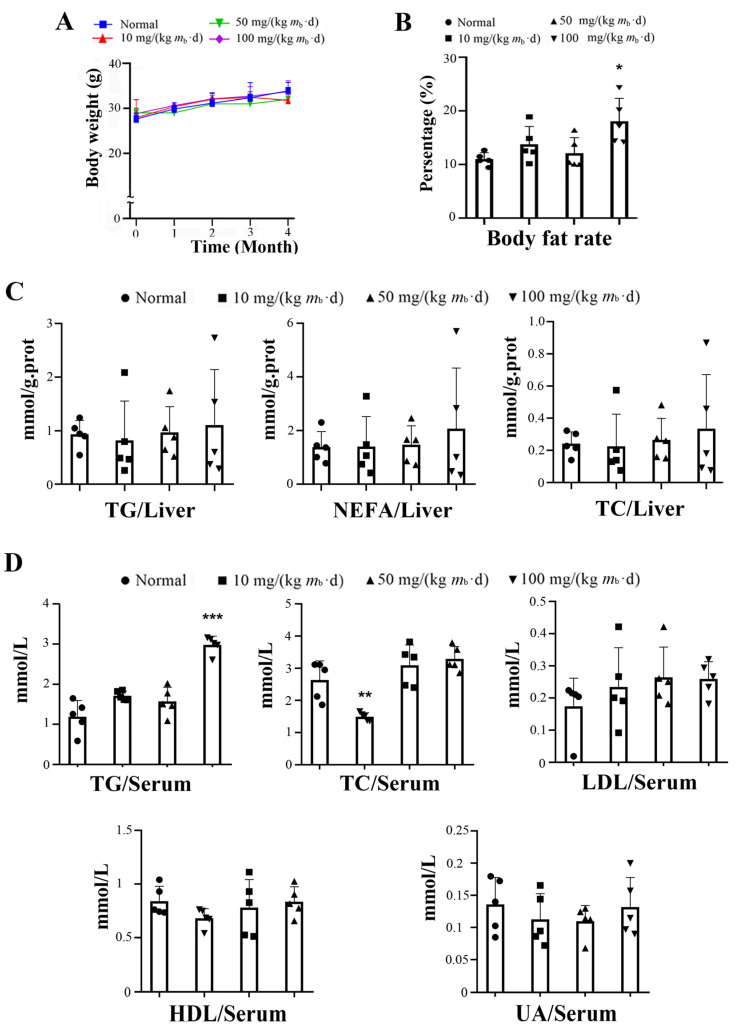
Effects on lipid metabolism in mice treated by various intake doses of IMP (*n* = 5). (**A**) Statistics of body weights measured at intervals of 1 month during IMP treatment. (**B**) Statistics of body fat rates measured at the end of the IMP treatment. (**C**) Statistics of the liver contents of TG, NEFA, and TC from each group of mice. (**D**) Statistics of the serum contents of TG, TC, LDL, HDL, and UA from each group of mice. Data are expressed as mean ± SD; compared with the control group, * *p* < 0.05, ** *p* < 0.01, *** *p* < 0.001.

**Figure 2 ijms-24-14588-f002:**
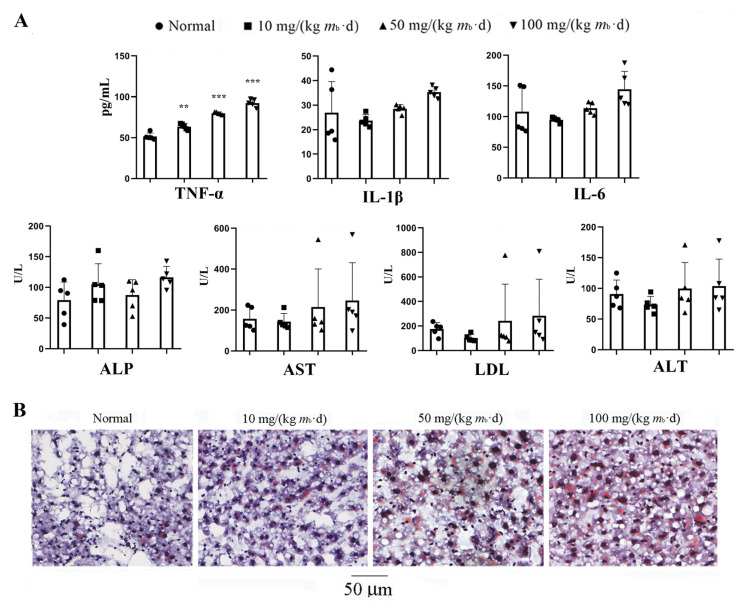
Effects on inflammation in mice treated with various intake doses of IMP (*n* = 5). (**A**) Changing trend in content of tumor necrosis factor-α (TNF-α), interleukin-1β(IL-1β), interleukin (IL-6), alkaline phosphatase (ALP), aspartate aminotransferase (AST), alanine aminotransferase (ALT), and lactate dehydrogenase (LDH) in the serum of mice, induced by various oral intake doses of IMP. Statistics of the data are expressed as mean ± SD; compared with the control group, ** *p*< 0.01, *** *p* < 0.001. (**B**) Histological morphology of liver tissue stained with Oil Red O. Each photo was obtained from a mouse whose TGs content was higher than that of other members in the same group.

**Figure 3 ijms-24-14588-f003:**
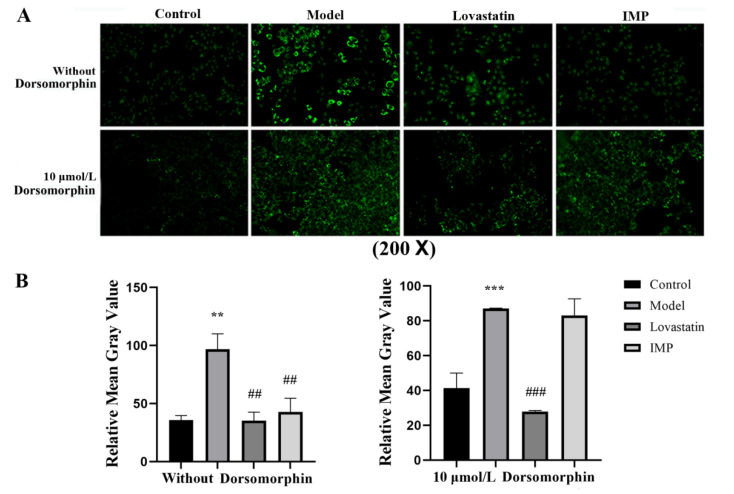
IMP lipid-lowering activity with and without AMPK inhibitor in in vitro assay (*n* = 5). (**A**) HepG2 cells in each of the groups were stained with Bodipy 493/503 solution and images of the cells were observed with a fluorescence microscope at 503 nm. (**B**) The intensity of green fluorescence represents the number of lipid droplets. Data are expressed as mean ± SD; compared with the control group, ** *p* < 0.01, *** *p* < 0.001; with model group, ## *p* < 0.01, ### *p* < 0.001.

**Figure 4 ijms-24-14588-f004:**
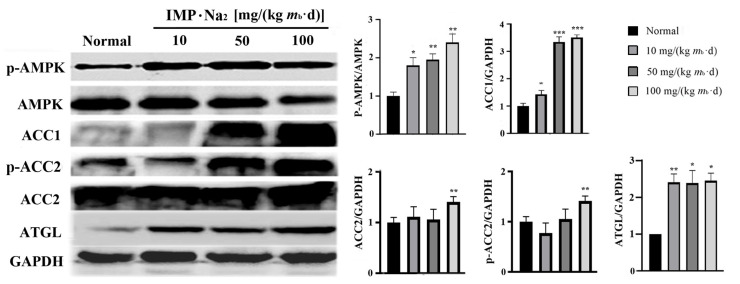
Relative expression levels of phosphorylated AMPK, acetyl-CoA carboxylase 1 (ACC1), acetyl-CoA carboxylase 2 (ACC2) as well as phosphorylated ACC2, and adipose triglyceride lipase (ATGL), in the liver of the mice in each group were determined using Western blot assay (*n* = 3). (Left) Images of Western blot assay. (Right) The relative expression levels of the each of the proteins are expressed as mean ± SD; compared with the control group, * *p* < 0.05, ** *p* < 0.01, *** *p* < 0.001.

**Figure 5 ijms-24-14588-f005:**
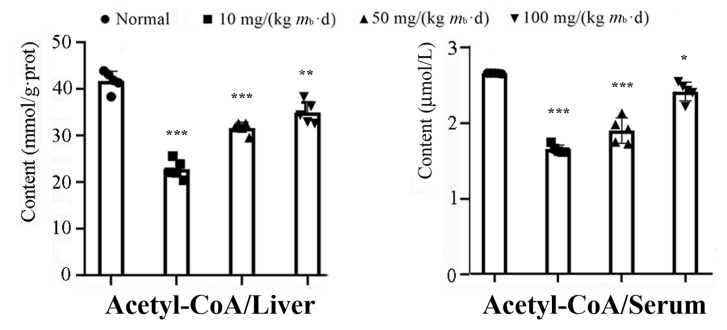
Content changes in acetyl-CoA in liver tissue and serum of mice, induced via various oral intake doses of IMP (*n* = 5). Data are expressed as mean ± SD; compared with the control group, * *p* < 0.05, ** *p* < 0.01, *** *p* < 0.001.

**Figure 6 ijms-24-14588-f006:**
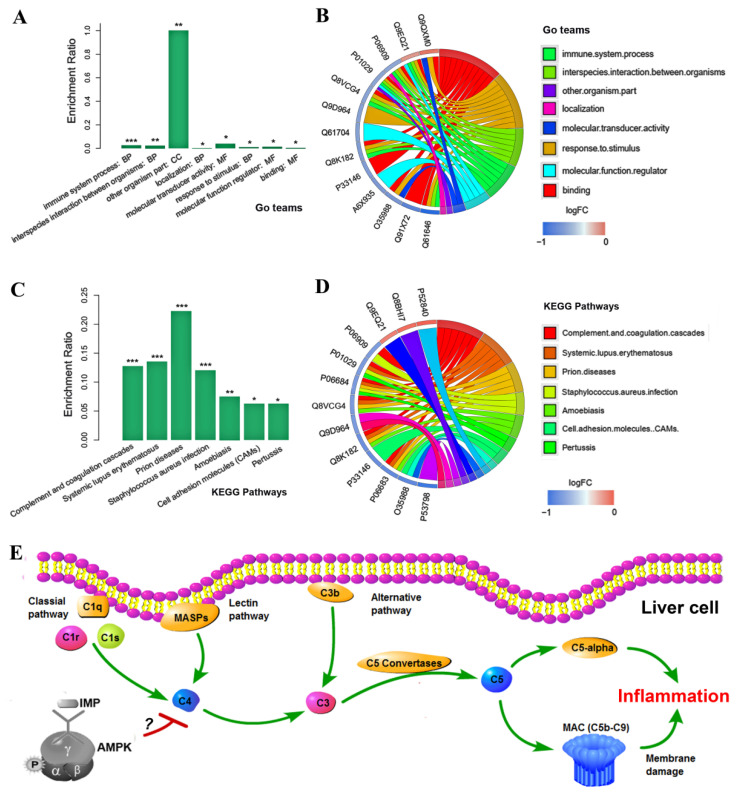
Results of bioinformatics analysis and pathway of IMP-induced inflammation. (**A**) Enriched GO functions and (**B**) proteins associated with these GO functions as well as (**C**) enriched KEGG pathways and (**D**) expression changed proteins associated with eash KEGG pathway Compared with the control group, * *p* < 0.05, ** *p* < 0.01, *** *p*< 0.001, (*n* = 3). (**E**) Diagram of IMP-induced inflammation pathway based on the up-regulated and down-regulated proteins in the complement system of mice. IMP-induced down-regulated proteins located in the complement system are shown in blue.

**Figure 7 ijms-24-14588-f007:**
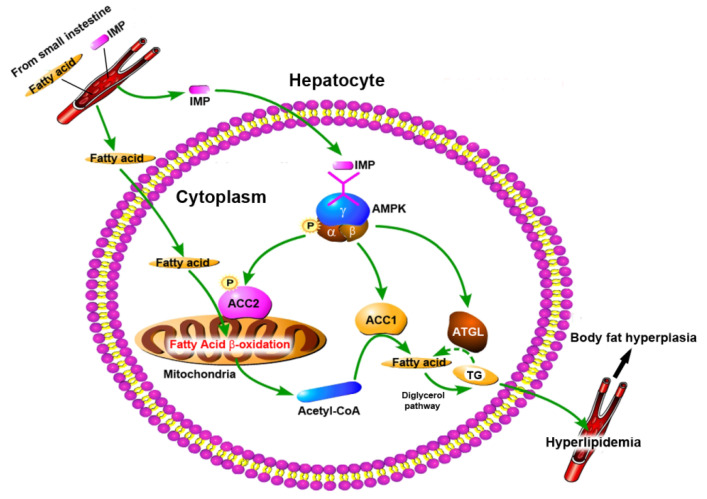
Diagram of pathway of IMP-induced hyperlipidemia in healthy mice.

**Table 1 ijms-24-14588-t001:** List of IMP-induced up-expressed proteins in the healthy mice hepatocytes.

Accession No.	Symbol	Description	Ratio	*p*-Value (*n* = 3)
Q9EQ21	HEPC	Hepcidin	1.20	0.0257
P52840	ST1A1	Sulfotransferase 1A1	1.26	0.0331
Q9QXM0	ABHD2	Mono-acylglycerol lipase ABHD2	1.30	0.0353
Q8BHI7	ELOV5	Elongation of very long chain fatty acids protein 5	1.24	0.0436

**Table 2 ijms-24-14588-t002:** List of IMP induced down-expressed proteins in the healthy mice hepatocytes.

Accession No.	Symbol	Description	Ratio	*p*-Value (*n* = 3)
Q8K182	C8A	Complement component C8 *α* chain	0.79	0.0024
A6X935	ITIH4	Inter α-trypsin inhibitor, heavy chain 4	0.78	0.0049
P06684	C5	Complement C5	0.81	0.0069
P06909	CFAH	Complement factor H	0.82	0.0081
P06683	C9	Complement component C9	0.78	0.0091
Q61646	HPT	Haptoglobin	0.60	0.0114
Q9ET22	DPP2	Dipeptidyl peptidase 2	0.81	0.0129
Q8K558	TRML1	Trem-like transcript 1 protein	0.81	0.0143
Q8VCG4	C8G	Complement component C8 *γ* chain	0.81	0.0170
Q9D1M7	FKB11	Peptidyl-prolyl cis-trans isomerase FKBP11	0.78	0.0230
P58044	IDI1	Isopentenyl-diphosphate Delta-isomerase 1	0.81	0.0264
P01029	C4B	Complement C4-B	0.81	0.0320
P53798	FDFT	Squalene synthase	0.72	0.0320
Q61704	ITIH3	Inter-α-trypsin inhibitor heavy chain H3	0.81	0.0341
O35988	SDC4	Syndecan-4	0.77	0.0346
Q91X72	HEMO	Hemopexin	0.70	0.0356
P33146	CAD15	Cadherin-15	0.78	0.0372
Q8R2E9	ERO1B	endoplasmic reticulum oxidoreductase 1-like protein β	0.79	0.0391
Q9D964	GATM	Glycine amidinotransferase, mitochondrial	0.81	0.0408
Q6P5D3	DHX57	Putative ATP-dependent RNA helicase DHX57	0.79	0.0434
P01642	KV5A9	Ig *κ* chain V-V region L7 (Fragment)	0.78	0.0440
Q9R112	SQOR	Sulfide:quinone oxidoreductase, mitochondrial	0.80	0.0465
P63168	DYL1	Dynein light chain 1, cytoplasmic	0.82	0.0485

## Data Availability

The data presented in this study are within the paper and Appendix A.

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
