# Peer review of "Oral Intake of Inosine 5′-Monophosphate in Mice Promotes the Absorption of Exogenous Fatty Acids and Their Conversion into Triglycerides though Enhancing the Phosphorylation of Adenosine 5′-Monophosphate-Activated Protein Kinase in the Liver, Leading to Lipohyperplasia"

_ijms, 2023, doi:10.3390/ijms241914588_

Round 1

Reviewer 1 Report

Opinion about manuscript entitiled „Oral intake of inosine 5'-monophosphate (IMP) in mice promotes the absorption of exogenous fatty acids and their conversion into triglycerides though enhancing the phosphorylation of AMPK in liver, leading to lipohyperplasia” sent to International Journal of Molecular Sciences (MDPI).

After the reading I am convinced that the text has got the merit and after some improvements it will be widely read by audience. It is interesting and contains line of valuable and useful information. I am working in the field and that text will help me in the future scientific activity.

1. What is the main question addressed by the research?

The authors tried to answer the question whether metabolic functions of C57BL/6J mice are affected by various doses of gavaged  (for 4 months) IMP Inosine 5’-monophosphate. The in vivo model has been chosen properly.

2. Do you consider the topic original or relevant in the field? Does it address a specific gap in the field?

Although the topic is not entirely original as the literature provides tons of knowledge about IMP as a metabolic agent, there is still a gap about specific mechanisms related to IMP-liver-disease and what are consequences of that status hepatic health. The experimental schema is to be admired.

3. What does it add to the subject area compared with other published material?

I suppose that the literature is quite rich in knowledge about the topic but the authors gave us the opportunity to see more details because the experimental schema is OK and it provides opportunity to answer “big” questions.

4. What specific improvements should the authors consider regarding the methodology? What further controls should be considered?

The construction of the study is very right.

5. Are the conclusions consistent with the evidence and arguments presented and do they address the main question posed?

The conclusions are right and justified by the main text. However, the authors must state that the mice number n=5 is not so high to draw final indisputable conclusions.

6. Are the references appropriate?

The references were used in an appropriate manner.

7. Please include any additional comments :

- line 40: My co-worker and I found that… - please use the Passive voice throughout the text.

- please add information about how 4 months-old mice at the start and 8-months old at the termination of the study should be considered – as mature or elderly?

- Figure 1 and others – some elements of the figures are vague. Please improve the resolution of pictures.

- The discussion section must be widen by a proper discussion with the findings of other researchers. Now, it is poor discussed with literature.

- line 333: fetal bovine serum (Gibco) – please provide information about Gibco Company (city country etc.). Check the whole text.

- the reference section must be checked: journal names are provided in different ways: abbreviated or not , with dots or not, etc.

- reference 19: ? Please translate.

Author Response

Responses to reviewer

RE: IJMS2592595

We appreciate deeply for your kind judgment to our manuscript (IJMS2592595).

We have carefully examined your comments and have responded to each of the comments mentioned in the letter as follows.

Comments:

Are the conclusions consistent with the evidence and arguments presented and do they address the main question posed?

The conclusions are right and justified by the main text. However, the authors must state that the mice number n=5 is not so high to draw final indisputable conclusions.

Reply: We agree your comments. We state that the mice number n=5 is not so high to draw final indisputable conclusions.

Based on our data, we can understand how to purine nucleotides such as IMP promote adipose accumulation to cause metabolic syndrome.

Please include any additional comments:

  1. - line 40: My co-worker and I found that… - please use the Passive voice throughout the text.

Reply: It had been revised. Please see the revised manuscript line, 42.

  1. - Please add information about how 4 months-old mice at the start and 8-months old at the termination of the study should be considered – as mature or elderly?

Reply: Based on the age relation between mice and humans, 4-8 months-old mice are equivalent to 25-35 years-old humans, which are young group and have not still reached middle age. Metabolic syndroms occurred in this age group were commonly not caused by aging, so 4-month-old mice treated by IMP at various dose evary day for 4 months can show the mechanism of IMP directly affecting the health of normal mammals.

  1. - Figure 1 and others – some elements of the figures are vague. Please improve the resolution of pictures.

Reply: It had been reivsed.

  1. The discussion section must be widen by a proper discussion with the findings of other researchers. Now, it is poor discussed with literature.

Reply: The discussion section was widened on the pathogenesis of hyperlipidemia based on the findings of other researchers, please see the revised manuscript.

  1. - line 333: fetal bovine serum (Gibco) – please provide information about Gibco Company (city country etc.). Check the whole text.

Reply: It had been reivsed.

  1. - the reference section must be checked: journal names are provided in different ways: abbreviated or not , with dots or not, etc.

Reply: It had been revised.

  1. - reference 19: ? Please translate.

Reply: It had been translated in to English.

We have carefully examined comments from all of reviews and revised the manuscript. We deeply appreciate your consideration of our manuscript again. We hope that this paper will be quickly published as soon as possible to guide the peoples to avoid health damage due to excessive IMP intake. If you have any enquiries, please don’t hesitate to contact me at the address below.

Yours sincerely,

Yong Xie 

Institute of Medicinal Plant Development, Chinese Academy of Medical Science & Peking Union Medical college, Malianwa North Road No.151, Haidian District, Beijing 100193, China

E-mail: yxie@implad.ac.cn

Tel: +86-10-57833280

Reviewer 2 Report

I congratulate the Authors on a very interesting paper entitled "Oral intake of inosine 5'-monophosphate (IMP) in mice promotes the absorption of exogenous fatty acids and their conversion into triglycerides though enhancing the phosphorylation of AMPK in liver, leading to lipohyperplasia". In my opinion, the work is prepared correctly, accurately, although it requires minor corrections. Below are my comments:

1. in the summary it is worth explaining the abbreviations - NAFLD, ACC!, ACC2, TG

2. it is worth highlighting (it is better to indicate in the text) the purpose of the work.

3. as in the abstract, abbreviations should be explained the first time they appear in the text (e.g. UA, TG, NEFA, TC, LDL, HDL, AST, ALT...)

4. Figure 3 (the picture in this figure is not labeled - what does it mean? what does DMSO mean?

Other than that, I have no major comments on the article.

No.

Author Response

Responses to reviewer

RE: IJMS2592595

We appreciate deeply for your kind judgment to our manuscript (IJMS2592595).

We have carefully examined your comments and have responded to each of the comments mentioned in the letter as follows.

Comments:

  1. in the summary it is worth explaining the abbreviations - NAFLD, ACC!, ACC2, TG

Reply: The abbreviations in this manuscript had been eaplained. Please see the   revised manuscript.

  1. it is worth highlighting (it is better to indicate in the text) the purpose of the work.

Reply: We had revised the abstract section and introduction section. Please see line 13 to line 15 and line

  1. as in the abstract, abbreviations should be explained the first time they appear in the text (e.g. UA, TG, NEFA, TC, LDL, HDL, AST, ALT...)

Reply: The abbreviations in this manuscript had been eaplained.

  1. Figure 3 (the picture in this figure is not labeled - what does it mean? what does DMSO mean?

Reply: Due to the solubility in water of lovastatin or dorsomorphin is low, and the addition of DMSO in cell culture can improve the solubility of lovastatin or dorsomorphin. In vitro lipid-lowering activity assay was carried out using a cell culture medium containing 0.1 (v/v) DMSO. Based on your comment, Figure 3 had been revised.

We have carefully examined comments from all of reviews and revised the manuscript. We deeply appreciate your consideration of our manuscript again. We deeply appreciate your consideration of our manuscript again. We hope that this paper will be quickly published as soon as possible to guide the peoples to avoid health damage due to excessive IMP intake. If you have any enquiries, please don’t hesitate to contact me at the address below.

Yours sincerely,

Yong Xie 

Institute of Medicinal Plant Development, Chinese Academy of Medical Science & Peking Union Medical college, Malianwa North Road No.151, Haidian District, Beijing 100193, China

E-mail: yxie@implad.ac.cn

Tel: +86-10-57833280

Reviewer 3 Report

The manuscript present interesting data about the intake of IMP and its effect on the lipid metabolism in mice.  The Abstract of the paper presents the main findings of the study  in a concise way. The Introduction also contains sufficient information about the reasons to conduct such a study. However (Line 40) the authors state : "Me and my co-worker found....". This is not an appropriate way to say, especially if there is a reference cited, usually it is said/ written " We found....", or "in a previous study... we found....". Furthermore, Line 56-59 say: "Judging from the normal life habits of humans, IMP is usually ingested from
food and food additive while diet, thus, we can consider that the metabolic functions of C57/BL6J mice effected by the gavage of IMP every day are consistent with those of intake of IMP from food in humans. " How was this found? Is there any proof, reference, please, cite.

The results are clearly presented and the graphic material is rich. The number of figures is adequate. The experimental design is appropriate, the methods applied are new and reliable. The conclusions need to be reformed. In my opinion they are too concise. However, the authors recommend some limits for human intake of the IMP. This is not correct in my opinion, and additional experiments with people from various age categories, weight, etc., are necessary to make sound recommendations.

Many language errors have been detected throughout the text. My advice is that authors address a native English speaker for the language of the paper.

Author Response

Responses to reviewer

RE: IJMS2592595

We appreciate deeply for your kind judgment to our manuscript (IJMS2592595).

We have carefully examined your comments and have responded to each of the comments mentioned in the letter as follows.

Comments:

  1. However (Line 40) the authors state : "Me and my co-worker found....". This is not an appropriate way to say, especially if there is a reference cited, usually it is said/ written " We found....", or "in a previous study... we found....".

Reply: It had been revised. Please see the revised manuscript, line 42.

  1. Furthermore, Line 56-59 say: "Judging from the normal life habits of humans, IMP is usually ingested fromfood and food additive while diet, thus, we can consider that the metabolic functions of C57/BL6J mice effected by the gavage of IMP every day are consistent with those of intake of IMP from food in humans. " How was this found? Is there any proof, reference, please, cite.

Reply:According to your comment, we had revised the "Judging from the normal life habits ……from food in humans. ". In the revised manuscript, reference 14 and 15 was cited to support that morden human developed a eating habits regularly. The two references had been added into the reference list.

  1. The results are clearly presented and the graphic material is rich. The number of figures is adequate. The experimental design is appropriate, the methods applied are new and reliable. The conclusions need to be reformed.In my opinion they are too concise. However, the authors recommend some limits for human intake of the IMP. This is not correct in my opinion, and additional experiments with people from various age categories, weight, etc., are necessary to make sound recommendations.

Reply: We had revised the conclusion section. Please see the revised manuscript.

Comments on the Quality of English Language

Many language errors have been detected throughout the text. My advice is that authors address a native English speaker for the language of the paper.

Reply: Our manuscript had been polished by native English speaker, all of the language errors had been corrected. 

We have carefully examined comments from all of reviews and revised the manuscript. We deeply appreciate your consideration of our manuscript again. We deeply appreciate your consideration of our manuscript again. We hope that this paper will be quickly published as soon as possible to guide the peoples to avoid health damage due to excessive IMP intake. If you have any enquiries, please don’t hesitate to contact me at the address below.

Yours sincerely,

Yong Xie 

Institute of Medicinal Plant Development, Chinese Academy of Medical Science & Peking Union Medical college, Malianwa North Road No.151, Haidian District, Beijing 100193, China

E-mail: yxie@implad.ac.cn

Tel: +86-10-57833280